# More direct attacks increase likelihood of goals in 2018- and 2022-Men's World Cup Soccer Finals

**Tim Taha**◉*, **Ilya Orlov**

Faculty of Kinesiology and Physical Education, University of Toronto, Toronto, Ontario, Canada

* t.taha@utoronto.ca

## Abstract

In soccer, attacking tactics can vary between elaborate, high passing play and play that involves very direct, straight-line action towards the towards the opponent's goal. It is of considerable interest to individuals involved which type of play is more effective in scoring given that goals are a rare event. We propose a geometric measure of directness (DIR) using the ratio between the straight-line distance from the point where possession begins to the centre of goal, and the total distance covered by the ball during that possession. Using 128 matches from the 2018- and 2022-Men's World Cup, we analyzed the influence of directness (DIR), speed of the ball traveling towards the goal (SPG) and the starting position of the possession (XPOS) on the likelihoods of shots and goals. A mixed-effect multivariate logistic regression model was used for both analyses. Following model simplification (AIC = 14579.7, $R^2$ = 0.279), the log odds of a shot resulting from a possession was significantly increased by XPOS ($\beta$ = 0.019, p < 0.0001), SPG ($\beta$ = 0.322, p < 0.0001) and a three-way interaction between DIR, XPOS and SPG ($\beta$ = 0.007, p < 0.0001). The likelihood of a shot was decreased by interactions between DIR and XPOS ($\beta$ = −0.024, p < 0.0001), DIR and SPG ($\beta$ = −0.587, p < 0.0001) and XPOS and SPG ($\beta$ = −0.003, p < 0.0001. The model for the likelihood of a goal (AIC = 1736.9, $R^2$ = 0.020) was simple with DIR being the only significant factor ($\beta$ = 1.009, p < 0.0001). The results suggest that to increase the likelihood of scoring goals, attacking tactics must be more direct.

## Introduction

Possession in football can be characterized as a series of actions undertaken by players of the attacking team to achieve a common aim. While ostensibly that common goal may be scoring a goal, possessions that result in goals are relatively rare [1]. As the only way to win a football match is to score at least one goal more than the opposition, the elements inherent in these rare possessions that result in goals become of high interest.

Discreet elements such as the number of passes within a possession were counted in early work and fewer passes were found to increase the likelihood of scoring [2]. Other early work divided the pitch into thirds where goals were found to be more likely to occur when a possession originated in the attacking third [3]. The results of the early work led to the suggestion that direct play consisting of a low number of passes and playing the ball into the attacking

**Data availability statement:** The data underlying the results presented in the study are

available from: https://github.com/statsbomb/open-data

**Funding:** The author(s) received no specific funding for this work.

**Competing interests:** The authors have declared that no competing interests exist.

zone as quickly as possible was most likely to result in increased goal scoring [3]. However, by the late 2000s, teams such as Barcelona and the Spanish Men's National Team had considerable success using a high possession type game. Yet, studies from that time confirmed the earlier findings. In a more fine-grained analysis, Tenga et al., showed what they termed as 'direct play' was more likely to result in possession within the scoring box (an area in front of the opposition's goal with a high probability of scoring) than 'possession play' when the defense was not able to establish its structure and was found to be 'unbalanced' [1,4]. Direct play attacks in Tenga et al., were noted for progressing relatively quickly in comparison to possession play attacks [1]. The definition of 'possession play' or 'direct play' was based mainly on the play of the defense. If the defense was more unbalanced or unorganized, then the play was considered 'direct' and if the defense was balanced and organized, they play was considered 'possession' [4]. This binary distinction for possession or direct play was not used by Collett [5]. Instead, time of possession was regressed with variables such as mean points per game. Top leagues in Europe that were studied collectively showed greater mean points per game with higher possession times [5]. However, when elite teams such as those reaching the knock-out stages of the UEFA Champions league were removed, the relationship between mean possession time and mean points per game was substantially reduced [5]. More recently, Taha and Ali found that the likelihood of scoring went down in possessions originating in the defensive one-third of the pitch with increasing possession times and increasing numbers of passes [6]. While this is not surprising when compared to previous findings, the teams analyzed included the World Cup winning Spanish and German Men's teams which were teams that played a possession game [6]. The results suggest that there is considerable variation in time of possession and other factors within a given possession even amongst teams that are considered to use elaborate attacks. Sarmento et al., suggested that simple notational analysis was not giving enough information or context and incorporated interviews with coaches and players to better understand differences in attacking elements between three top level European professional teams [7]. Their inclusion of qualitive data suggests that notational measures such as pass counts or zones where events during a possession occur are not enough to give a complete picture of what elements comprise a successful possession. To better understand these previous findings different means of analysis must be utilized to better understand the possessions that result in scoring in either direct play or possession based tactical systems.

Spatial analysis in football has made great leaps forward allowing better understanding of player positioning. From initial roots in simply dividing the pitch into regions (e.g., attacking or defensive) and counting player actions (e.g., pass or tackle) in those regions [2,3], to locating players and the ball using xy-coordinates based on time and pitch measurements [8,9] and associating those times and locations with actions has allowed researchers to get novel insights into areas such as game related measures of fitness [10] and integration of the physical and tactical actions of players [11]. Fernandez and Bornn were able to develop an innovative model of player influence using automated quantitative analysis of player position and movement on the pitch [12]. This allowed them to show how teams created space through player and ball movement [12,13]. The data for this type of research has been previously the domain of professional clubs and private organizations due primarily to the complexity and cost of collecting and processing the information [14]. Recently, companies in this area have publicly released data sets that allow spatial analysis which democratizes research in this area. Data from these sets is rich giving locations of players when they first touch the ball, whether they dribble, pass, or shoot and who on the field gets the next touch [15]. This allows tracking of a possession, giving both notational data such number of passes within a possession and continuous data such as locations on the pitch and the time at which this location has occurred, allowing calculation of speed of the ball and players and the distances covered by each.

In the current study, we conducted a fine-grained spatial analysis to better understand what elements are in a successful possession. Given contradictory evidence from previous work where elite high possession teams were found to gain more points per game [5], yet still show possessions that are shorter in time and have fewer passes were more likely to result in goals [6], we were interested in looking at the path of the ball during possessions. We hypothesized that possessions where the ball traveled in more direct paths to the opposing goal were more likely to result in scoring.

To test our hypothesis, we developed a metric of the directness of play (detailed in the methods section) in a possession. We analyzed 128 games from the 2018- and 2022-Men's World Cup as this was the most recent large data sets available and therefore the analysis would reflect the most up to date trends in the international game.

## Methods

### Data

Data from 128 matches during the 2018- and 2022-Men's World Cups used in this study was collected by Statsbomb.com and is freely available through their Github page [15]. Statsbomb uses a team of five people to collect data from live video of games. Two collectors tag players and locations of player actions, one collector notes major events and a fourth collector who assigns information and details about each event or player action. The fifth team member acts as a reviewer to limit errors made in the process. Additionally, a proprietary algorithm is used to tag players involved in sequential actions to also limit errors and speed up the rate of collection. Within the available data is time and pitch locations for actions undertaken by players in possession of the ball. The data is coded with an event (e.g., pass, reception, carry, shot), the time during the match the event occurred and the xy coordinates of each of the events corresponding to a soccer pitch that has a length of 120 units and a width of 80 units. For the 128 matches, 461641 events were recorded. For this study, we used possessions that consisted of more than a single event. For example, a pass by one player that is received by a teammate would be included while a single touch that is turned over to the opponent would not be included. Possessions that were the result of corner kicks and other set-pieces were excluded. For our analysis, we included 370319 events that were part of 22661 possessions. Table 1 gives the extracted data for a possession that resulted in a goal by Argentina in the 2022 World Cup Final between Argentina and France. Fig 1 maps out the data from Table 1.

**Table 1. Extracted data showing a possession by Argentina resulting in a goal during the 2022 Men's World Cup Final between Argentina and France.**

| Home team | Away team | Match id | Event index | Timestamp | Possession | Possession team | Type of possession | Location | Shot outcome | Duration |
|---|---|---|---|---|---|---|---|---|---|---|
| Argentina | France | 3869685 | 1166 | 35:12.9 | 52 | Argentina | Pass | [25.9, 71.8] | NA | 1.466588 |
| Argentina | France | 3869685 | 1167 | 35:14.4 | 52 | Argentina | Ball Receipt | [41.6, 59.2] | NA | NA |
| Argentina | France | 3869685 | 1168 | 35:14.4 | 52 | Argentina | Pass | [41.8, 59.0] | NA | 0.986942 |
| Argentina | France | 3869685 | 1169 | 35:15.3 | 52 | Argentina | Ball Receipt | [53.1, 61.8] | NA | NA |
| Argentina | France | 3869685 | 1170 | 35:15.3 | 52 | Argentina | Carry | [53.1, 61.8] | NA | 0.466769 |
| Argentina | France | 3869685 | 1171 | 35:15.8 | 52 | Argentina | Pass | [52.9, 61.8] | NA | 1.429727 |
| Argentina | France | 3869685 | 1172 | 35:17.2 | 52 | Argentina | Ball Receipt | [62.5, 68.2] | NA | NA |
| Argentina | France | 3869685 | 1173 | 35:17.2 | 52 | Argentina | Carry | [62.5, 68.2] | NA | 0.070904 |
| Argentina | France | 3869685 | 1174 | 35:17.3 | 52 | Argentina | Pass | [63.2, 68.2] | NA | 3.651671 |
| Argentina | France | 3869685 | 1175 | 35:21.0 | 52 | Argentina | Ball Receipt | [99.3, 54.9] | NA | NA |
| Argentina | France | 3869685 | 1176 | 35:21.0 | 52 | Argentina | Pass | [99.3, 54.9] | NA | 1.661003 |
| Argentina | France | 3869685 | 1177 | 35:22.6 | 52 | Argentina | Ball Receipt | [111.8, 32.1] | NA | NA |
| Argentina | France | 3869685 | 1178 | 35:22.6 | 52 | Argentina | Shot | [111.8, 32.1] | Goal | 0.466153 |

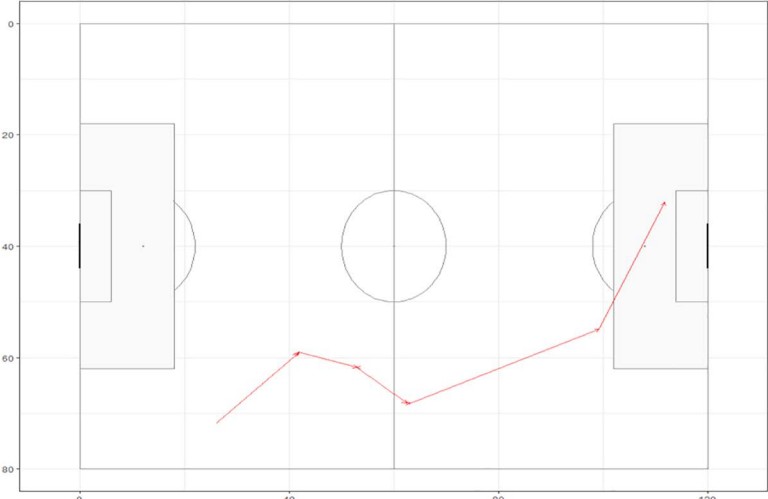

**Fig 1. Argentinian possession that resulted in a goal during the 2022 Men's World Cup Final between Argentina and France.** Direction of attack for Argentina is from left to right.

The study was approved by the ethics committee of the Faculty of Kinesiology and Physical Education at the University of Toronto. The committee waived the need for consent.

## Data analysis

The data analysis was completed using the time and the xy-coordinates of all the events within a given team possession. Using the xy-coordinates, we calculated the total distance covered by the ball within a possession. For example, using the values from the first pass in Table 1 (location 25.9, 71.8 to location 41.6, 59.2), we calculated a distance of 20.1 units using the Pythagorean formula. For the entire possession each of those individual distances were summed. In the example using the Argentinian possession in Table 1, a total distance of 109.0 units was found. We also calculated the direct distance of the possession along a line passing through the location of the beginning of the possession to the centre of the goal on the goal line (xy-coordinates 120, 40). The distance that the ball travelled along that line during the possession was calculated. In the possession in Table 1, the calculated direct distance was found to be 94.1 units (Fig 2). Using the direct distance and the total distance we were able to calculate the ratio between the two values. We termed this ratio the directness of the possession.

$$directness = \frac{direct\ distance}{total\ distance} \tag{1}$$

A value of 1 would mean that the ball was played directly to the goal and values approaching zero suggest that the ball is moving with greater amounts of lateral or backwards movement in relation to the goal. In the example of the Argentinian possession in Table 1 the directness ratio was found to be 0.86.

In addition, we also calculated the speed of the ball traveling along the same line used in the directness measure that passes through the initial event of the possession and the centre of the goal line. The time difference between the beginning and the end of the possession and the direct distance was used to the calculate the speed of the possession towards the goal. In the possession used in Table 1, the speed towards the goal is calculated to be 9.7 units/second.

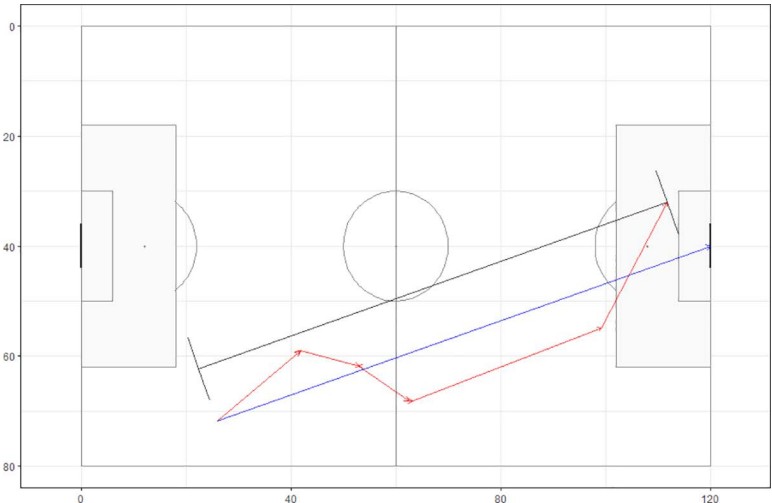

**Fig 2. Argentinian possession that resulted in a goal during the 2022 Men's World Cup Final between Argentina and France.** The blue line is the direct line to the centre of the goal from the beginning of the possession and the black line shows the direct distance along the blue line that the possession covered.

Finally, we determined the location of the beginning of each possession using the x-coordinate. This allowed us to see where the possession originated in terms of the length of the pitch.

## Statistical modeling

We created two statistical models to investigate the contributions of directness (DIR), the speed of the ball travelling to the goal (SPG) and the starting position (XPOS) to the likelihood of a possession resulting in a shot and the likelihood of a shot resulting in a goal. We utilized mixed-effects multivariate logistic regression on the suggestion of Tenga et al., due to autocorrelation within the data because of the playing style of a given team within a given World Cup [1]. As the outcome variables of shot or no shot and goal or no goal are binomial, we used a binomial distribution to calculate the log-odds of a shot or goal occurring. We began with fully saturated models that included main effects (DIR, SPG and XPOS) and their interactions. The models were then simplified by removing the non-significant interactions and main effects and re-calculating Akaike's 'An Information Criterion' (AIC). The final models were chosen based on a low AIC and significant main effects and interactions. Goodness of fit ($R^2$) of each of the models was calculated using the methods of Nakagawa and Schielzeth [16] for generalized-linear mixed methods models. The model for the log-odds of a shot took the final form:

$$Log\ odds\ shots = \beta_{intercept} + \beta_1(DIR) + \beta_2(XPOS) + \beta_3(SPG) + \beta_4(DIR\ x\ XPOS)$$
$$+ \beta_5(DIR\ x\ SPG) + \beta_6(XPOS\ x\ SPG) + \beta_7(DIR\ x\ XPOS\ x\ SPG) + \in \quad (2)$$

$$\beta_{intercept} = \beta_{mean\,intercept} + \beta_{team|world\,cup\,intercept} \quad (2a)$$

where $\beta_{1,2,3,4,5,6,7}$ are the coefficients for the main effects (DIR, XPOS, and SPG) and the significant interactions. As the teams at a given World Cup (2018 or 2022) would have common tactics or style of play, the intercepts (random effects) were allowed to vary to accommodate the potential autocorrelation. $\in$ is an error term. The model for the likelihood of a goal given a shot is simpler having only one main fixed effect taking the form:

$$Log\,odds\,goals\,given\,a\,shot = \beta_{intercept} + \beta_1(DIR) + \in \qquad (3)$$

$$\beta_{intercept} = \beta_{mean\,intercept} + \beta_{team|world\,cup\,intercept} \qquad (3a)$$

Neither of the models had overdispersion.

## Results

Of the 2018- and 2022-Men's World Cup event and possession data that was analyzed, 22661 possessions were included in the study. Of those possessions, 2878 resulted in a shot and of those shots, 372 resulted in goals.

The first analysis examined the effect of directness (DIR), the speed of the ball travelling to the goal (SPG) and the starting position (XPOS) on the log odds likelihood of a possession resulting in a shot. Random effects intercepts in both log odds and probability for each team at the respective World Cups are shown in Table 2 in columns 1 and 3. In terms of fixed effects, increasing the speed of the ball traveling to the goal (SPG, Estimate: 0.322, p < 0.0001) and gaining the possession of the ball closer to the opposition's goal (XPOS, Estimate: 0.019, p <0.0001) increased the log odds of a shot. Significant interactions were also found (Table 3) between the three main effects. DIR interacted with XPOS such that more direct play in possessions gained closer to the opposition's goal resulted in a small reduction in log odds of a shot (Estimate: −0.024, p < 0.0001). More direct play interacts with higher speed towards the goal to result in a reduction in log odds (Estimate: −0.587, p < 0.0001). The final two-way interaction between higher speed towards the opposition's goal and possessions gained closer to the opposition's goal resulted in a small decrease in log odds of a shot (Estimate: −0.003, p < 0.0001). The three-way interaction between all three of the main effects resulted in a small positive effect on the log odds of a shot being taken during a possession (Estimate: 0.007, p < 0.0001). The results of the main effects and interactions with three different distances to the opposing goal can be seen in Fig 3.

The second analysis, which began with a fully saturated model that examined the effects of directness (DIR), the speed of the ball travelling to the goal (SPG) and the starting position (XPOS) on the log odds of a goal given a possession that resulted in a shot, simplified to a single significant main effect. Directness of play was the only fixed effect that impacted the log odds of a possession that ended with a shot resulting in a goal (Estimate: 1.009, p < 0.0001; Table 4). More direct play resulted in a positive effect on the log odds of a goal. Random effects for each team for the respective World Cup are given in columns 2 and 4 in Table 2

## Discussion

In the current study, we used possession and position data from 128 games from the 2018- and 2022-Men's World Cup to understand the effect of ball's direction, speed of travel and starting position in a possession on the likelihood of a shot or goal. Like previous work, we used logistic mixed effects models to account for the differing playing styles of teams during a given World Cup competition [1,6]. For shots, the results were somewhat complex, with an increased log odds of a shot with higher speeds towards the goal (0.322), gaining possession of the ball closer to the opposition's goal (0.019), and a three-way interaction of all main effects (0.007). The likelihood of a shot was decreased by significant two-way interactions between directness and gaining possession closer to the opposition's goal (−0.024), directness and higher speeds towards the goal (−0.587), and higher speeds towards the goal and possessions

**Table 2. Random effect estimated log odds intercepts and base probabilities for possessions that result in a shot and possessions that result in a goal for each team at the 2018, and 2012 World Cup Finals.**

| | Base Log Odds for a Possession resulting in a shot | | Base Log Odds for a shot resulting in a goal | | Base Probability for a possession to result in a shot | | Base Probability for a shot resulting in a goal | |
|---|---|---|---|---|---|---|---|---|
| Team | 2018 | 2022 | 2018 | 2022 | 2018 | 2022 | 2018 | 2022 |
| Argentina | −2.89 | −2.84 | −2.43 | −2.42 | 0.11 | 0.12 | 0.18 | 0.18 |
| Australia | −2.98 | −3.06 | −2.64 | −2.48 | 0.10 | 0.09 | 0.14 | 0.17 |
| Belgium | −2.67 | −2.84 | −2.30 | −2.61 | 0.14 | 0.12 | 0.20 | 0.15 |
| Brazil | −2.45 | −2.63 | −2.55 | −2.53 | 0.17 | 0.14 | 0.16 | 0.16 |
| Colombia | −2.95 | | −2.41 | | 0.10 | | 0.18 | |
| Cameroon | | −2.94 | | −2.41 | | 0.11 | | 0.18 |
| Canada | | −2.94 | | −2.61 | | 0.11 | | 0.15 |
| Costa Rica | −2.94 | −3.10 | −2.56 | −2.44 | 0.11 | 0.09 | 0.15 | 0.17 |
| Croatia | −2.79 | −3.05 | −2.44 | −2.43 | 0.12 | 0.09 | 0.18 | 0.18 |
| Denmark | −2.92 | −2.97 | −2.54 | −2.60 | 0.11 | 0.10 | 0.16 | 0.15 |
| Ecuador | | −3.01 | | −2.46 | | 0.10 | | 0.17 |
| Egypt | −2.80 | | −2.62 | | 0.12 | | 0.15 | |
| England | −2.91 | −2.83 | −2.48 | −2.23 | 0.11 | 0.12 | 0.17 | 0.21 |
| France | −2.88 | −2.71 | −2.41 | −2.27 | 0.11 | 0.13 | 0.18 | 0.21 |
| Germany | −2.49 | −2.62 | −2.70 | −2.53 | 0.17 | 0.15 | 0.14 | 0.16 |
| Ghana | | −3.00 | | −2.34 | | 0.10 | | 0.19 |
| Iceland | −2.85 | | −2.60 | | 0.12 | | 0.15 | |
| Iran | −2.93 | −2.87 | −2.61 | −2.49 | 0.11 | 0.11 | 0.15 | 0.17 |
| Japan | −2.83 | −2.91 | −2.46 | −2.42 | 0.12 | 0.11 | 0.17 | 0.18 |
| Mexico | −2.66 | −2.88 | −2.68 | −2.62 | 0.14 | 0.11 | 0.14 | 0.14 |
| Morocco | −2.82 | −2.91 | −2.63 | −2.52 | 0.12 | 0.11 | 0.14 | 0.16 |
| Netherlands | | −3.22 | | −2.18 | | 0.08 | | 0.23 |
| Nigeria | −2.81 | | −2.56 | | 0.12 | | 0.15 | |
| Panama | −2.97 | | −2.54 | | 0.10 | | 0.16 | |
| Peru | −2.79 | | −2.56 | | 0.12 | | 0.15 | |
| Poland | −2.98 | −3.04 | −2.55 | −2.59 | 0.10 | 0.10 | 0.16 | 0.15 |
| Portugal | −2.79 | −2.83 | −2.51 | −2.33 | 0.12 | 0.12 | 0.16 | 0.19 |
| Qatar | | −3.05 | | −2.55 | | 0.09 | | 0.16 |
| Russia | −2.96 | | −2.33 | | 0.10 | | 0.19 | |
| Saudi Arabia | −2.92 | −2.97 | −2.60 | −2.47 | 0.11 | 0.10 | 0.15 | 0.17 |
| Senegal | −2.90 | −2.81 | −2.52 | −2.52 | 0.11 | 0.12 | 0.16 | 0.16 |
| Serbia | −2.80 | −2.88 | −2.62 | −2.40 | 0.12 | 0.11 | 0.15 | 0.18 |
| South Korea | −2.84 | −2.83 | −2.51 | −2.44 | 0.12 | 0.12 | 0.16 | 0.17 |
| Spain | −2.72 | −2.97 | −2.48 | −2.28 | 0.13 | 0.10 | 0.17 | 0.20 |
| Sweden | −2.81 | | −2.57 | | 0.12 | | 0.15 | |
| Switzerland | −2.85 | −3.02 | −2.48 | −2.40 | 0.12 | 0.10 | 0.17 | 0.18 |
| Tunisia | −2.82 | −2.96 | −2.47 | −2.61 | 0.12 | 0.10 | 0.17 | 0.15 |
| United States | | −2.85 | | −2.55 | | 0.12 | | 0.16 |
| Uruguay | −2.87 | −2.82 | −2.49 | −2.57 | 0.11 | 0.12 | 0.17 | 0.15 |
| Wales | | −3.07 | | −2.62 | | 0.09 | | 0.15 |

gained closer to the opposition's goal (−0.003). The likelihood of goals resulted in a simple model where the log odds of a goal occurring within a possession significantly increased with more directness within the possession (1.009).

**Table 3. Estimated log odds, standard errors, confidence intervals and p-values for a possession resulting in a shot.**

| | Estimate | Standard Error | 5% Confidence Interval | 95% Confidence Interval | p–value |
|---|---|---|---|---|---|
| Intercept | −2.875 | 0.124 | −3.119 | −2.632 | <0.0001 |
| Directness | 0.450 | 0.316 | −0.169 | 1.069 | 0.155 |
| Possession Start | 0.019 | 0.001 | 0.016 | 0.021 | <0.0001 |
| Direct Speed to Goal | 0.322 | 0.058 | 0.209 | 0.436 | <0.0001 |
| Directness X Possession Start | −0.024 | 0.005 | −0.033 | −0.015 | <0.0001 |
| Directness X Direct Speed to Goal | −0.587 | 0.070 | −0.725 | −0.450 | <0.0001 |
| Possession Start X Direct Speed to Goal | −0.003 | 0.001 | −0.005 | −0.002 | <0.0001 |
| Directness X Possession Start X Direct Speed to Goal | 0.007 | 0.001 | 0.006 | 0.009 | <0.0001 |

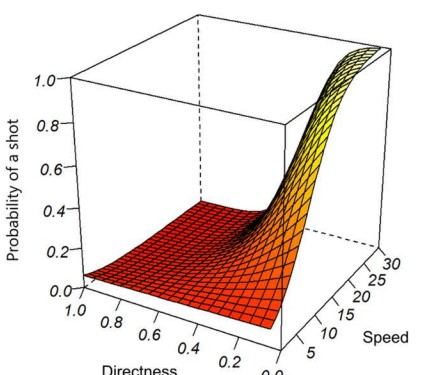 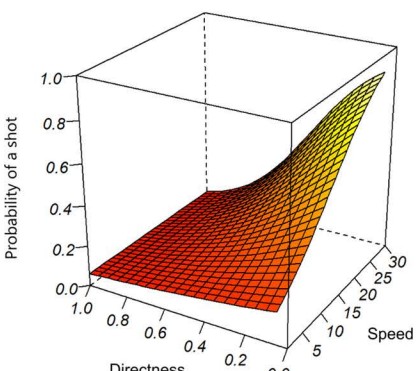 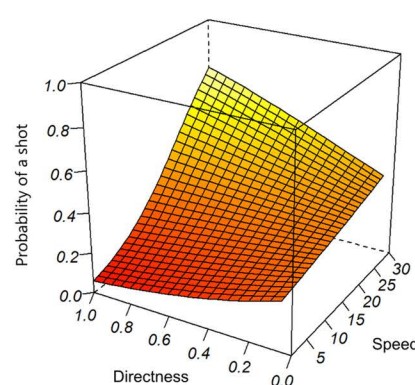

**Fig 3. The estimated contributions of directness and speed of the ball towards the opposing goal to the probability of shot from within the defensive 18-yard box** (left graph, x = 15), the centre line of the pitch (middle graph, x = 60) and outside the attacking 18-yard box (right graph, x = 95).

**Table 4. Estimated log odds, standard errors, confidence intervals and p-values for possessions that result in a goal.**

| | Estimate | Standard Error | 5% Confidence Interval | 95% Confidence Interval | p–value |
|---|---|---|---|---|---|
| Intercept | −2.496 | 0.110 | −2.712 | −2.281 | <0.0001 |
| Directness | 1.009 | 0.234 | 0.551 | 1.468 | <0.0001 |

Previous analyses using spatial data have been classified in several systematic reviews [8,11,17,18]. Using the classification of Sarmento et al. (2018) [9], the current study can be considered an analysis of sequential patterns of group behaviour during matches. Other studies that have a similar classification have found that teams have certain temporal patterns [19] of how an offensive possession progresses towards the opponent's goal. For example, a team may initially move the ball to the centre of the pitch through the middle and then attempt attacks by using the pitch width in the attacking zones. In the current study, using a mixed effects approach we can show that despite differences in these temporal patterns between teams, there are common elements of which directness is suggested to be the most significant in terms of goal scoring. Other studies have also looked at the likelihood of a possession resulting in a shot or goal. The difference between those studies and the current work lies mainly in how possessions are tracked. Due to the lack of spatial tracking, previous studies have relied heavily on dividing the pitch into zones [3,4,17,20–22]. With the data provided

by Statsbomb, we were able to track the locations of player touches on the ball. Given this data, we developed the directness measure as a simple mathematical tool to assess the degree of direct play within a possession. Previous work, such as Tenga et al., defined direct play based on the levels of unbalance within the defense during a possession [1,4,23]. For example, a direct possession was one that utilized or created defensive imbalances by the opponent. Furthermore, the direct play would begin with a pass or dribble that penetrates the defense [4]. Possession play differed in their definition in that the penetration would occur late in the possession [4]. These types of definitions are useful but rely on many factors, such as defensive positioning and movement of the offense, to determine the type of possession. Many of these factors can only be determined by observation rather than measurement. The directness measure, that we demonstrate in this study, removes some of the potential ambiguity from the concept of direct or possession play by relying solely on measured values.

In comparing the results of the present study to previous work, we find that there are numerous points of agreement. In the studies that looked at types of offensive play and goal scoring, there are findings where direct or counterattacking play is significantly more likely to result in goals [1,4,17,24]. Our findings that possessions that move in a straighter line to the goal are more likely to score goals suggest that tactics such as counterattacks which rely on moving the ball towards the goal as directly and quickly as possible will increase the likelihood of scoring. Rico-Gonzales et al., in their review, came in a similar conclusion for attacking in general and suggested that attacking play must be "long" with attacking players positioned in a rectangle whose longer side parallels the length of the field [25]. Fewer studies examined shots, with most looking at scoring opportunities [17,24] or score box possession [14,20] as measures of successful possessions. Like the finding by Sarmento et al., that shorter possessions were more likely to result in offensive success [17], we found that higher speeds towards the goal were more likely to result in a shot on goal. We also were able to show that possessions that originated closer to the goal were more likely to result in a shot which was a common finding dating back to early studies in football offense [3]. Our finding that the interaction between increasing directness and the possession beginning closer to the opposition goal was like the findings by Lago-Ballesteros et al [20]. Using a multiple logistic regression, they found that direct attacks originating in the pre-offensive zone had a lower likelihood of success than direct or elaborate attacks originating in the pre-defensive zone [20].

The negative interactions between direct play and possessions originating close to the goal are intriguing. While this may be due to fitting of the model, it could also be explained by greater opponent pressure on the shooter closer to the opposition's goal [21] that would be higher in a direct play that is begun close to the goal. Potentially, possessions that originate in the offensive corners of the pitch close to the goal line may require movement of the ball away from the centre of the goal (mainly towards the defensive goal) to achieve possession in an area of high dangerousity [26]. In our model, this would mean a lower amount of directness within those possessions to the opposition's goal. This may also account for our finding that the likelihood of a shot decreased with higher speeds towards the goal and the possession originating closer to the opposition's goal line.

Higher speeds of movement of the ball towards the opposition goal with more directness resulted in a lower likelihood of a shot. Again, this may be due to fit of the model. Alternatively, higher speeds of movement of the ball which could be due to a high difficulty of the pass [27], can make it more difficult to control the ball increasing the possibility of a turnover prior to a shot being made.

There are limitations to the current study. Critically, due to the data only giving the beginning and end locations of a pass or dribble, we must assume that the path of the player or ball is always in a straight line. There would be some curvature with a pass especially if the ball is

traveling through the air. In terms of dribbling, we would expect that the path of the player varies from a straight line. For the most part, this would result in longer measured distances than we found, lowering the directness value on average. Future work with data that tracks player and ball locations continuously will clarify if this under-estimation is significant. A second limitation is due to using statistical models to describe the describe the data. In the current study, the complexity of the shots model may stem from the fitting of the data. Some of the interaction terms seem contradictory to the main effects. For example, the main effects of the starting position and the speed of the ball towards the goal have a positive effect on the log odds of a shot during a possession. However, the interaction term between the two was found to lower the log odds. Given that, it is important to note that a statistical model represents the best fit of the data [28] which can result in findings that have apparent contradictions such as those found in the shots model in the current study.

Our findings suggest that teams adopt tactics that move the ball directly to the opposition's goal to maximize the probability of goal scoring. All football actions that players undertake within such a system must be completed with a view to move the ball quickly in as straight a line as possible to the opponent's goal. However, game theory would suggest that doing this on every possession gained would be detrimental as defenses would adapt their structure to counteract the direct tactics minimizing the effectiveness of direct play [29]. Mixed tactics of randomly varying possessions between direct play and more elaborate slower, passing based possessions would be suggested by game theory [29]. From a player's perspective this would mean that they would require the skills to play in both systems as different sets of football actions are necessary in either tactic. From a team point of view all players on the team gaining possession of the ball must quickly arrive on a common tactic without giving too much information to the opposing team. If some players play directly and others do not, then confusion may arise which can potentially increase the likelihood of a turnover.

Future work must integrate the current understanding of player influence [13,14,30] to allow better understanding of why more direct play increases the likelihood of goals. There may be certain defensive conditions that offensive players observe and exploit that allow more direct play to be more successful. Understanding those conditions or situations will help defenses develop tactics to minimize scoring while offenses will look to either passively or actively [12] create those situations. Context of the game, which has been noted to effect tactical play [31], must be also incorporated to gain a better understanding of the importance of direct play in goal scoring [12–14,30].

## Conclusion

At the 2018- and 2022-men's World Cup, possessions that were gained closer to the opposition's goal and the ball was moved at a higher speed were more likely to result in a shot. This higher shot likelihood was tempered by possessions that had more directness and were gained close to the opponent's goal and those that had high levels of directness and ball speed towards the opponent's goal. More importantly, as goals decide the outcomes of games as opposed to shots, more directness to the opponent's goal during a possession resulted in a significantly higher likelihood of scoring.

## Author contributions

**Conceptualization:** Tim Taha, Ilya Orlov.

**Data curation:** Tim Taha.

**Formal analysis:** Tim Taha.

**Investigation:** Tim Taha, Ilya Orlov.

**Methodology:** Tim Taha, Ilya Orlov.

**Software:** Tim Taha.

**Writing – original draft:** Tim Taha.

**Writing – review & editing:** Tim Taha, Ilya Orlov.

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
