## [Decision Letter · Decision Letter 0]

8 Oct 2024

PONE-D-24-35283More direct attacks increase likelihood of goals in 2018- and 2022-Men’s World Cup Soccer FinalsPLOS ONE

Dear Dr. Taha,

Thank you for submitting your manuscript to PLOS ONE. After careful consideration, we feel that it has merit but does not fully meet PLOS ONE’s publication criteria as it currently stands. Therefore, we invite you to submit a revised version of the manuscript that addresses the points raised during the review process.

We look forward to receiving your revised manuscript.

Kind regards,

Emiliano Cè

Academic Editor

PLOS ONE

Journal Requirements:

2. Please note that your Data Availability Statement is currently missing the repository name. If your manuscript is accepted for publication, you will be asked to provide these details on a very short timeline. We therefore suggest that you provide this information now, though we will not hold up the peer review process if you are unable.

Additional Editor Comments:

Dear Authors,

your manuscript has been revised by one expert in the field that reported several minor points you should consider during the revision process. 

Reviewers' comments:

Reviewer's Responses to Questions

**Comments to the Author**

1. Is the manuscript technically sound, and do the data support the conclusions?

Reviewer #1: Yes

2. Has the statistical analysis been performed appropriately and rigorously? 

Reviewer #1: Yes

3. Have the authors made all data underlying the findings in their manuscript fully available?

Reviewer #1: Yes

4. Is the manuscript presented in an intelligible fashion and written in standard English?

Reviewer #1: Yes

5. Review Comments to the Author

Reviewer #1: General comments

This manuscript proposed a geometric measure of directness (DIR) using the ratio between the straight-line distance from the point where possession begins to the centre of goal, and the total distance covered by the ball during that possession. The sample was powerful with 128 matches from the 2018-25 and 2022-Men’s World Cup, as well as the quite key indicators performance like directness (DIR), speed of the ball traveling towards the goal (SPG) and the starting position of the possession (XPOS). Also, the research is based on an interesting research problem based on the attacking tactics can vary between elaborate, high passing play and play that involves very direct, straight-line action towards the towards the opponent’s goal. However, some sections need improvement to clarify some critical points of the research.

Specific comments

Abstract: It should add the goodness criteria used to validate the accuracy of the mixed-effect multivariate logistic regression model. Also, replace ‘Estimate’ with the corresponding mathematical symbol.

Introduction: The introduction is very well structured, however lacks some context as to the time-series and tracking data that the study will use to measure fine-grained spatial analysis to better understand what elements are in a successful possession (please, see: https://peerj.com/articles/14381/).

Methods: The Statsbomb.com is a database with very interesting soccer information. However, it would be possible to add some more information about the origin of the data. How was it collected? What instruments and software were used? From the point of view of validity and internal consistency, this information is very important from a scientific point of view. The authors could even question the software organization about it. Also, it is necessary to define the goodness criteria for the statistical analysis on which the analysis is based. Please see: https://www.mdpi.com/2075-4663/10/8/121

Results: The results are robust and well-founded. Only figure 1 and 2 seem to me to present the other possibilities, because describing only one possibility of more direct attacks increasing likelihood for such a large sample seems too ambitious:

Discussion: The discussion is well-focused, but I recommend that you expand the comparison of the studies with other reference studies in the field of tactical analysis in soccer. In order to expand the references to 35 references, I recommend the following linked works:

- https://hrcak.srce.hr/318343

- https://link.springer.com/article/10.1007/s40279-019-01194-7

- https://www.tandfonline.com/doi/full/10.1080/02640414.2014.898852

- https://link.springer.com/article/10.1007/s40279-017-0836-6

-https://www.termedia.pl/Reference-values-for-collective-tactical-behaviours-based-on-positional-data-in-professional-football-matches-a-systematic-review,78,43095,0,1.html

https://www.sciencedirect.com/science/article/pii/S0960077920301120

- https://journals.plos.org/plosone/article?id=10.1371/journal.pone.0247067

6. PLOS authors have the option to publish the peer review history of their article (what does this mean?). If published, this will include your full peer review and any attached files.

Reviewer #1: **Yes: **José Eduardo Teixeira

---

## [Author Response · Author response to Decision Letter 1]

9 Nov 2024

Dear Prof. Cè,

Thank you for giving us the opportunity to respond to the reviewer. The reviewer had excellent and informed comments and suggestions that we incorporated into the revised manuscript. The changes in response to the reviewer’s suggestions have strengthened the paper by making it clearer to understand. Our responses can be found in the “Response to reviewers” file. All the changes have been made in the new version of the “Manuscript” file.

Again, thank you for the opportunity to respond to the reviewer.

Warmest Regards

Tim Taha, PhD

Faculty of Kinesiology and Physical Education

University of Toronto

Toronto, Ontario, Canada

General Comments

Also, the research is based on an interesting research problem based on the attacking tactics can vary between elaborate, high passing play and play that involves very direct, straight-line action towards the towards the opponent’s goal. However, some sections need improvement to clarify some critical points of the research.

Thank you for your clear summary and the understanding of our over-arching research problem. The current study represents a small but important portion of the research problem. The reviewer’s suggestions provide some excellent means to clarify the current study.

Abstract

It should add the goodness criteria used to validate the accuracy of the mixed-effect multivariate logistic regression model. Also, replace ‘Estimate’ with the corresponding mathematical symbol.

Thank you for the useful suggestion. We added AIC and R2 to the abstract. We also replaced Estimate with the symbols

Line 28 (Ln 28 in Manuscript)

Following model simplification (AIC = 14579.7, R2 = 0.279), the log odds of a shot resulting from a possession was significantly increased by XPOS (β¬ = 0.019, p <0.0001), SPG (β = 0.322, p < 0.0001) and a three-way interaction between DIR, XPOS and SPG (β = 0.007, p <0.0001). The likelihood of a shot was decreased by interactions between DIR and XPOS (β¬ = -0.024, p < 0.0001), DIR and SPG (β = -0.587, p < 0.0001) and XPOS and SPG (β¬ = - 0.003, p < 0.0001. The model for the likelihood of a goal (AIC = 1736.9, R2 = 0.020) was simple with DIR being the only significant factor (β¬ = 1.009, p < 0.0001).

Introduction

The introduction is very well structured, however lacks some context as to the time-series and tracking data that the study will use to measure fine-grained spatial analysis to better understand what elements are in a successful possession (please, see: https://peerj.com/articles/14381/).

Thank you for the positive comment. We worked very hard to try to structure the introduction to be concise yet give the reader sufficient information to understand the over-arching research question. The suggestion to add some more context to spatial and time-series data is helpful as it will give a reader a brief introduction or review of the state of the art. The papers recommended by the reviewer will also allow a reader to get a further understanding of the power of spatial and time series data. We included the following:

Line 86 (Ln 79 in Manuscript)

Spatial analysis in football has made great leaps forward allowing better understanding of player positioning. From initial roots in simply dividing the pitch into regions (e.g., attacking or defensive) and counting player actions (e.g., pass or tackle) in those regions[1,2], to locating players and the ball using xy-coordinates based on time and pitch measurements [3,4] and associating those times and locations with actions has allowed researchers to get novel insights into areas such as game related measures of fitness[5] and integration of the physical and tactical actions of players[6].

Methods

The Statsbomb.com is a database with very interesting soccer information. However, it would be possible to add some more information about the origin of the data. How was it collected? What instruments and software were used? From the point of view of validity and internal consistency, this information is very important from a scientific point of view. The authors could even question the software organization about it.

We found the Statsbomb.com open database to be very interesting as well. While the information on their methods of collection can be found on the website, it is not clearly labelled. We agree with you that we should present it to make it easier for readers. Rather than search through a website, readers can get a clear overview of the methods of collection. We included the following paragraph:

Line 121 (Ln 111 in Manuscript)

Statsbomb uses a team of five people to collect data from live video of games. Two collectors tag players and locations of player actions, one collector notes major events and a fourth collector who assigns information and details about each event or player action. The fifth team member acts as a reviewer to limit errors made in the process. Additionally, a proprietary algorithm is used to tag players involved in sequential actions to also limit errors and speed up the rate of collection.

Also, it is necessary to define the goodness criteria for the statistical analysis on which the analysis is based. Please see: https://www.mdpi.com/2075-4663/10/8/121.

We thank you for this suggestion. We have included a goodness of fit measure (R2) using the methods of Nakagawa and Schielzeth[7] as we utilized generalized-linear mixed methods models in our study. We added the following in the methods:

Line 186 (Ln 178 in Manuscript)

Goodness of fit (R2) of each of the models was calculated using the methods of Nakagawa and Schielzeth[7] for generalized-linear mixed methods models.

We also reported AIC and R2 for each model in the results section.

Results

The results are robust and well-founded. Only figure 1 and 2 seem to me to present the other possibilities, because describing only one possibility of more direct attacks increasing likelihood for such a large sample seems too ambitious

Thank you for your positive comment on the nature of the results. We were surprised as well that the likelihood of goals was the only factor significantly linked to the directness of the play. We had expected interactions like what we had found with shots. Our hope is that others can build upon this finding with more in-depth work. We believe that including the areas of influence of the defense will result in a better understanding of why direct paths to the goal are effective. It may be that defensive positioning may open these paths which result in easy straightforward passage of the ball to the goal. The current data set limited us in terms of this kind of analysis, but we believe that it would be a promising avenue of research.

Conclusion

The discussion is well-focused, but I recommend that you expand the comparison of the studies with other reference studies in the field of tactical analysis in soccer. In order to expand the references to 35 references, I recommend the following linked works:

- https://hrcak.srce.hr/318343

- https://link.springer.com/article/10.1007/s40279-019-01194-7

- https://www.tandfonline.com/doi/full/10.1080/02640414.2014.898852

- https://link.springer.com/article/10.1007/s40279-017-0836-6

-https://www.termedia.pl/Reference-values-for-collective-tactical-behaviours-based-on-positional-data-in-professional-football-matches-a-systematic-review,78,43095,0,1.html

https://www.sciencedirect.com/science/article/pii/S0960077920301120

- https://journals.plos.org/plosone/article?id=10.1371/journal.pone.0247067

Thank you for this useful list of papers. We thoroughly reviewed the suggestions and included a number of changes in the discussion to expand the comparison between the present study and previous work.

Line 250

Previous analyses using spatial data have been classified in several systematic reviews[3,8,9,6]. Using the classification of Sarmento et al. (2018)[4], the current study can be considered an analysis of sequential patterns of group behaviour during matches. Other studies that have a similar classification have found that teams have certain temporal patterns[10] of how an offensive possession progresses towards the opponent’s goal. For example, a team may initially move the ball to the centre of the pitch through the middle and then attempt attacks by using the pitch width in the attacking zones. In the current study, using a mixed effects approach we can show that despite differences in these temporal patterns between teams, there are common elements of which directness is suggested to be the most significant in terms of goal scoring. Other studies have also looked at the likelihood of a possession resulting in a shot or goal. The difference between those studies and the current work lies mainly in how possessions are tracked.

Line 288 (Ln 278 in manuscript)

Our findings that possessions that move in a straighter line to the goal are more likely to score goals suggest that tactics such as counterattacks which rely on moving the ball towards the goal as directly and quickly as possible will increase the likelihood of scoring. Rico-Gonzales et al., in their review, came in a similar conclusion for attacking in general and suggested that attacking play must be “long” with attacking players positioned in a rectangle whose longer side parallels the length of the field [11].

Line 350 (Ln 337 in manuscript)

Future work must integrate the current understanding of player influence[12-14] to allow better understanding of why more direct play increases the likelihood of goals. There may be certain defensive conditions that offensive players observe and exploit that allow more direct play to be more successful. Understanding those conditions or situations will help defenses develop tactics to minimize scoring while offenses will look to either passively or actively [8] create those situations. Context of the game, which has been noted to effect tactical play [15], must be also incorporated to gain a better understanding of the importance of direct play in goal scoring.

References

1. Bate R. Football chance: Tactics and strategy. In: Reilly T, Lees A, Murphy WJ, Davids K, editors. Science and Football (Routledge Revivals): Proceedings of the first World Congress of Science and Football, Liverpool, 13-17th April 1987. Florence: Taylor and Francis; 1988. pp. 363–375.

2. Reep C, Benjamin B. Skill and Chance in Association Football. Journal of the Royal Statistical Society. Series A. General. 1968;131: 581–585. doi: 10.2307/2343726.

3. Sarmento H, Marcelino R, Anguera MT, Campanico J, Matos N, LeitÃo JC. Match analysis in football: a systematic review. Journal of sports sciences. 2014;32: 1831–1843. doi: 10.1080/02640414.2014.898852.

4. Sarmento H, Clemente FM, Araújo D, Davids K, McRobert A, Figueiredo A. What Performance Analysts Need to Know About Research Trends in Association Football (2012–2016): A Systematic Review. Sports Med. 2018;48: 799–836. doi: 10.1007/s40279-017-0836-6.

5. de Araújo Teixeria JE, Branquinho L, Leal M, Marinho DA, Ferraz R, Barbosa TM, et al. Modeling the major influencing factor on match running performance during the in-season phase in a Portuguese professional football team. Sports. 2022;10: 1–9. doi: 10.3390/sports10080121.

6. de Araújo Teixeria JE, Miguel Forte P, Ferraz R, Branquinho L, Silva AJ, Monteiro AM, et al. Integrating physical and tactical factors in football using positional data: a systematic review. PeerJ (San Francisco, CA). 2022;10: 1–32. doi: 10.7717/peerj.14381.

7. Nakagawa S, Schielzeth H, O'Hara RB. A general and simple method for obtaining R2 from generalized linear mixed‐effects models. Methods in Ecology and Evolution. 2013;4: 133–142. doi: 10.1111/j.2041-210x.2012.00261.x.

8. Sarmento H, Figueiredo A, Lago-Peñas C, Milanovic Z, Barbosa A, Tadeu P, et al. Influence of Tactical and Situational Variables on Offensive Sequences During Elite Football Matches. Journal of strength and conditioning research. 2018;32: 2331–2339. doi: 10.1519/JSC.0000000000002147.

9. Low B, Coutinho D, Gonçalves B, Rein R, Memmert D, Sampaio J. A Systematic Review of Collective Tactical Behaviours in Football Using Positional Data. Sports Med. 2020;50: 343–385. doi: 10.1007/s40279-019-01194-7.

10. Camerino OF, Chaverri J, Anguera MT, Jonsson GK. Dynamics of the game in soccer: Detection of T-patterns. European journal of sport science. 2012;12: 216–224. doi: 10.1080/17461391.2011.566362.

11. Rico-González M, Pino-Ortega J, Castellano J, Oliva-Lozano JM, Los Arcos A. Reference values for collective tactical behaviours based on positional data in professional football matches: a systematic review. Biology of Sport. 2022;39: 110–114. doi: 10.5114/biolsport.2021.102921.

12. Caetano FG, Barbon Junior S, Torres RdS, Cunha SA, Ruffino PRC, Martins LEB, et al. Football player dominant region determined by a novel model based on instantaneous kinematics variables. Scientific reports. 2021;11: 18209. doi: 10.1038/s41598-021-97537-4.

13. Martens F, Dick U, Brefeld U. Space and Control in Soccer. Frontiers in sports and active living. 2021;3: 676179. doi: 10.3389/fspor.2021.676179.

14. Memmert D, Lemmink KAPM, Sampaio J. Current Approaches to Tactical Performance Analyses in Soccer Using Position Data. Sports Med. 2017;47: 1–10. doi: 10.1007/s40279-016-0562-5.

15. Fernandez J, Bornn L. Wide open spaces: A statistical technique for measuring space creation in professional soccer. . 2018.

16. Praça GM, Lima BB, Bredt SdGT, Sousa RBE, Clemente FM, de Andrade AGP. Influence of Match Status on Players’ Prominence and Teams’ Network Properties During 2018 FIFA World Cup. Frontiers in psychology. 2019;10: 695. doi: 10.3389/fpsyg.2019.00695.

---

## [Decision Letter · Decision Letter 1]

14 Nov 2024

More direct attacks increase likelihood of goals in 2018- and 2022-Men’s World Cup Soccer Finals

PONE-D-24-35283R1

Dear Dr. Taha,

We’re pleased to inform you that your manuscript has been judged scientifically suitable for publication and will be formally accepted for publication once it meets all outstanding technical requirements.

Kind regards,

Emiliano Cè, Ph.D.

Academic Editor

PLOS ONE

Additional Editor Comments (optional):

Reviewers' comments:

Reviewer's Responses to Questions

**Comments to the Author**

1. If the authors have adequately addressed your comments raised in a previous round of review and you feel that this manuscript is now acceptable for publication, you may indicate that here to bypass the “Comments to the Author” section, enter your conflict of interest statement in the “Confidential to Editor” section, and submit your "Accept" recommendation.

Reviewer #1: All comments have been addressed

2. Is the manuscript technically sound, and do the data support the conclusions?

Reviewer #1: Yes

3. Has the statistical analysis been performed appropriately and rigorously? 

Reviewer #1: Yes

4. Have the authors made all data underlying the findings in their manuscript fully available?

Reviewer #1: Yes

5. Is the manuscript presented in an intelligible fashion and written in standard English?

Reviewer #1: Yes

6. Review Comments to the Author

Reviewer #1: Dear authors,

After the first round of revisions, we can see that the authors have substantially improved the entire manuscript and are endeavouring to respond constructively to all comments.

I therefore recommend accepting the manuscript in its present form. Congratulations on the excellent and pertinent research.

Best regards

7. PLOS authors have the option to publish the peer review history of their article (what does this mean?). If published, this will include your full peer review and any attached files.

Reviewer #1: **Yes: **José Eduardo Teixeira

---

## [Editor Report · Acceptance letter]

PONE-D-24-35283R1

PLOS ONE

Dear Dr. Taha,

I'm pleased to inform you that your manuscript has been deemed suitable for publication in PLOS ONE. Congratulations! Your manuscript is now being handed over to our production team.

Kind regards,

on behalf of

Prof. Emiliano Cè

Academic Editor

PLOS ONE